**Data Availability Statement:** Data are available from a public data repository https://www.openicpsr.org/openicpsr/project/171481/version/V1/view Project Citation: Dou, Yikai, Fan,

# Impact of the COVID-19 pandemic on psychological distress and biological rhythm in China's general population: A path analysis model

Yikai Dou[1], Huanhuan Fan[1], Xiao Yang[1], Yue Du[1], Yu Wang[1], Min Wang[1], Zijian Zhang[1], Xiongwei Qi[1], Yuling Luo[1], Ruiqing Luo[1], Xiaohong Ma 📷[1,2,3]*

1 Psychiatric Laboratory and Mental Health Center, West China Hospital of Sichuan University, Chengdu, Sichuan, China, 2 Psychiatric Laboratory and Mental Health Center, The State Key Laboratory of Biotherapy, West China Hospital of Sichuan University, Chengdu, Sichuan, China, 3 West China Brain Research Center, West China Hospital of Sichuan University, Chengdu, Sichuan, China

* maxiaohong@scu.edu.cn

## Abstract

### Objective

When facing major emergency public accidents, men and women may react differently. Our research aimed to assess the influence of gender difference on social support, information preference, biological rhythm, psychological distress, and the possible interaction among these factors during the COVID-19 pandemic.

### Methods

In this cross-sectional study, 3,237 respondents aged 12 years and older finished the online survey. Levels of social support, information preference, biological rhythm, and psychological distress were assessed using validated scales. A path analysis was conducted to explore possible associations among these variables.

### Results

The path analysis indicated that women with high levels of social support had a lower possibility of biological rhythm disorders and lower levels of somatization symptoms of psychological distress during the COVID-19 pandemic. The influence of social support on somatization symptoms was exerted via biological rhythm. Women tended to believe both negative and positive information, while men preferred more extreme information.

### Conclusion

Our results highlighted gender difference in study variables during the COVID-19 pandemic and the importance of social support in alleviating psychological distress and biological rhythm disorders. Moreover, we confirmed that information preference differed significantly by somatization symptoms of psychological distress, suggesting extra efforts to provide

Huanhuan, Yang, Xiao, Du, Yue, Wang, Yu, Wang, Min, . . . Ma, Xiaohong. Impact of the COVID-19 pandemic on psychological distress and biological rhythm in China's general population :A cross-sectional dataset. Ann Arbor, MI: Inter-university Consortium for Political and Social Research [distributor], 2022-05-27. https://doi.org/10.3886/E171481V1.

**Funding:** The author received no specific funding for this work.

**Competing interests:** The authors have declared that no competing interests exist.

more individualized epidemic information. Longitudinal research is required to further explore casual inferences.

## Introduction

Since the outbreak of the novel coronavirus disease 2019 (COVID-19) in late December, 2019, it has spread rapidly throughout almost all regions in the world [1]. The latest data form WHO's official website showed that this severe respiratory infectious disease has infected more than 500 million people and caused over 6 million to die (by early May 2022). COVID-19 presents an urgent and vital threat to global public health and social economy [2,3]. In many countries and regions, governments asked residents to reduce unnecessary outdoor activities, and shopping malls and public transportations were also closed to avoid intimate contact. Social distancing, self-isolation, and travel restrictions have led to downsize or closure of businesses as well as a reduced workforce across all economic sectors, causing many job losses and family income losses [4]. Even the Olympic Games that are held every four years had to be postponed. As an internationally concerned public health emergency, the COVID-19 pandemic has been influencing our regular lifestyle greatly and has a wide range of adverse psychological impacts on the general population [5–8].

Previous studies have shown that individuals may go through fear of being infected or even of death themselves, feeling hopeless or helpless and even ashamed once been infected [9]. A survey based on seven middle income countries in the general population showed high risk factors on mental health during the COVID-19 pandemic which related to single or separated status, high educated level, and age < 30 years [10]. Meanwhile, home quarantine can also cause high prevalence of symptoms of psychological distress such as insomnia, stress, emotional disturbance, and other psychological disorders [11]. Therefore, individual social support during home quarantine and accurate, timely and effective epidemic information is vital for the general public. The elderly, the single, the separated, and those who also live alone for various reasons have to face the horrible infectious disease without family members' company, which has caused a rising level of mental issues like anxiety, stress, and depression among such special population [12]. Insufficient medical supplies such as face masks and disinfectants at the beginning of the COVID-19 pandemic increased fear and uncertainty brought about by this severe viral infection [13].

Furthermore, social media using unreliable sources usually provide ambiguous epidemic information, and information overload may cause psychological distress in turn [14]. For example, while facemask wearing is a positive precaution, its use brings about social stigma, arousing mixed opinion and contradictory messaging from the media, which all lead to public fear and confusion [15]. Effective risk information communication among people can reduce negative psychological responses and strong social support may play a role [16]. In addition, information preference can be essential in helping shape the public's risk perception and has been reported to be influenced by people's gender, age, social status, etc. [17,18] Therefore, information preference should be considered when we analyze the underlying influencing factors of risk perceptions of infectious diseases such as COVID-19.

Besides social support and epidemic information preference, biological rhythm is another significant factor. Travel restrictions or home quarantine disturbs the circadian rhythms. Staying up late, getting up late, and lying in bed during non-sleeping time all decrease activity and

meal frequency, causing rhythm disorders in eating, sleeping, social activities, and aggravating people's physical and psychological distress in the meantime.

Therefore, we conducted this study during the COVID-19 pandemic in an attempt to identify a possible relationship of gender difference with social support, biological rhythm, information preference, and psychological distress. This is the first study to examine all these factors together in China's general population during the COVID-19 pandemic.

## Methods

### Study design and participants

A cross-sectional online survey was conducted via Chinese social applications (apps) WeChat and Weibo, the Chinese equivalent of Twitter, in China's mainland between 26 February, 2020 and 2 March, 2020. During this period most people were still isolated at home because of the COVID-19 pandemic. Participants would be excluded if they were under 12 years old or not living in China's mainland. This survey contained demographic information such as age, gender, education level, and social status, and took approximately 10-15 minutes for each participant to complete. Other vital information including social support, biological rhythm, media information preference, and psychological distress was also assessed. Informed consent was acquired before each participant decided to take this survey. For juvenile participants, informed consent was obtained from their parents or guardians. To protect the privacy of participants, all collected information was anonymous. This research was approved by the Ethics Committee of West China Hospital of Sichuan University (No.2020-178).

### Measure instruments

**Brief symptom inventory-18 (BSI-18).** BSI-18 is a self-report symptoms checklist, which is commonly used to evaluate psychological distress of respondents in the past one week [19]. It contains 18 items and can be divided into three subscales (somatization, depression, and anxiety). Scores of each item in this five-point Likert scale range from 0 (not at all) to 4 (very much). The total score of BSI-18 is also called "global severity index (GSI)". The Cronbach's alpha equals to 0.98, 0.94, 0.93, and 0.95 for GSI, somatization, depression, and anxiety, respectively, suggesting a good internal consistency reliability for our research sample. The Chinese version of BSI-18 has been used among China's patients and general population [20–22].

**Social Support Rating Scale (SSRS).** SSRS was used for the measurement of social support. It has been widely applied in different psychological studies; and its Chinese version was developed by Professor Xiao in 1998 [23]. SSRS consists of 10 items; and 3 dimensions of social support were evaluated, namely, subjective support (4 items), objective support (3 items), and support utilization (3 items). Scores of three subscales were simply added up, generating a social support total score ranging from 12 to 66. High scores demonstrate a higher level of social support received by the respondents [24,25]. In our research sample, the Cronbach's alpha of total support scores was 0.62, indicating a moderate reliability.

**Biological Rhythm Interview of Assessment in Neuropsychiatry (BRIAN).** RIAN was applied to assess the degree of biological rhythm dysregulation. This four-point scale contains 21 items. Four primary domains of rhythm disturbance, involving sleep (5 items), social rhythm (5 items), activity (4 items), and eating pattern (4 items) were evaluated. Another domain referring to chronotype was not taken into consideration in the total BRIAN score [26]. Higher total scores signify strong disturbance of biological rhythm. Previous studies show that BRIAN has good psychometric properties in patients with mood disorder or in general school students [27,28]. The scale has been translated into different versions. The Cronbach's alpha of total BRIAN scores in our sample was 0.95, indicating a good reliability [29].

**Media information preference.** Respondents' attitude toward media information was measured using one question: "Which kind of information do you usually pay attention to?" Two choices were provided: 1) Either negative media information or positive media information; and 2) Both negative and positive media information. This question was designed based on some previous researches which aimed to reflect the preference of different respondents for the magnanimity of media information on cellphone social apps or television [14,18,30,31].

## Statistical analysis

Data analysis was performed using Stata/SE 15.1 software. First, for continuous variables such as age and scores of psychological distress, *t* test was used to assess the statistical significance between men and women; for categorical variables between men and women, $X^2$ test was used to describe the constituent ratio of education level and media information preference, etc. Second, correlations between gender, media information preference, social support total scores, somatization scores, depression scores, anxiety scores, global severity index, and BRAIN total scores were calculated using Spearman's rank correlation coefficients. Finally, aiming to explore the overall relationship among multiple variables, we constructed a structural equation model (SEM) and applied path analysis to test the relationship among interrelated study variables in a hypothesized model. In our SEM, somatization scores were modeled as outcome variables, while gender was modeled as an observed variable; and social support total scores, media information preference, and biological rhythm were modeled as mediators. SEM estimated both the direct and indirect effects one variable had on the outcome variable. Several indices were used to determine whether the hypothesized model fit the observed data. The chi-square value was the original fit index for structural equation models. An acceptable model means p > 0.05 in the chi-square. However, some previous studies show that the chi-square test is so sensitive to sample size that it always rejects the SEM, especially when large samples are used [32]. Thus, several alternative fit indices were included in our study. Absolute fit indices such as the Root Mean Square Error of Approximation (RMSEA), the Standardized Root Mean Square Residual (SRMR), and the Goodness of Fit Index (GFI) were chosen to evaluate the structural model. It would be considered as a good model if RMSEA <0.08, SRMR <0.08, and GFI >0.90 [32–34]. Besides, incremental fit indices such as Tucker Lewis Index (TLI) and Comparative Fit Index (CFI) were also proposed. Values above 0.90 for TLI and CFI were considered an acceptable fit. Statistical significance was accepted at p < 0.05.

## Results

### Description of the sample

A total of 3,246 respondents registered in our questionnaire. In the end, 9 were excluded because they were below 12 years (n=8) or did not live in China's mainland (n=1). Their social-demographic information is shown in Table 1. Men (n=1,277) and women (n=1,960) differed significantly in residence ($X^2$=6.87), marital status ($X^2$=8.64), and information preference ($X^2$=9.92); the corresponding p values were 0.0090, 0.0030, and 0.0020, respectively. The two groups did not differ significantly in age (t=-1.23; p=0.2162) or education level ($X^2$=7.58; p=0.0560). Social support total scores, psychological distress scores, and BRIAN total scores were compared (Table 1). The mean scores of social support were obviously higher in women than in men (p=0.0008). Psychological distress scores differed significantly in somatization scores (p<0.0001) and global severity index (p=0.0088) between the two groups. The two groups did not differ significantly in depression scores, anxiety scores, or BRIAN total scores. (S1 Table shows the effect size of studying variables in Table 1).

**Table 1. Gender difference in social-demographic information, psychological distress, social support, biological rhythm, and media information preference.**

| | Men (N=1277) | Women (N=1960) | T/X$^2$ Value | p Value |
|---|---|---|---|---|
| *Age* | 30.70±9.36 | 31.12±9.56 | -1.23 | 0.2162 |
| *Education Level* | | | | |
| Middle School or Below | 122 (9.55%) | 136 (6.94%) | 7.58 | 0.0560 |
| High School | 148 (11.59%) | 221 (11.27%) | | |
| Bachelor's degree | 771 (60.38%) | 1221 (62.30%) | | |
| Master's degree or Above | 236 (18.48%) | 382 (19.49%) | | |
| *Residence* | | | | |
| Urban Areas | 768 (60.14%) | 1268 (64.69%) | 6.87 | 0.0090 |
| Rural Areas | 509 (39.86%) | 692 (35.31%) | | |
| *Marital Status* | | | | |
| Unmarried | 661 (51.76%) | 911 (46.48%) | 8.64 | 0.0030 |
| Married | 616 (48.24%) | 1049 (53.52%) | | |
| *Information Preference* | | | | |
| Positive or Negative Info. | 614 (48.08%) | 832 (42.45%) | 9.92 | 0.0020 |
| Both of the Above | 663 (51.92%) | 1128 (57.55%) | | |
| *Social Support Total Scores* | 37.57±8,26 | 38.54±7.89 | -3.36 | 0.0008 |
| *Somatization Scores* | 9.98±6.21 | 9.13±5.19 | 4.20 | <0.0001 |
| *Depression Scores* | 10.84±6.34 | 10.44±5.62 | 1.88 | 0.0606 |
| *Anxiety Scores* | 10.4±6.29 | 10.06±5.71 | 1.57 | 0.1170 |
| *Global Severity Index* | 31.22±18.37 | 29.64±15.66 | 2.62 | 0.0088 |
| *BRIAN Total Scores* | 31.27±12.58 | 32.07±11.54 | -1.86 | 0.0633 |

## Correlations among study variables

The correlations between study variables were studied (Table 2). Women had a positive correlation with media information preference (r=0.0554), social support scores (r=0.0546), and BRIAN total scores (r=0.0517). Meanwhile, Women had a negative correlation with somatization scores (r=-.00380). Besides, media information preference had a negative correlation with

**Table 2. Spearman correlations among study variables (N=3237).**

| Variables | 1 | 2 | 3 | 4 | 5 | 6 | 7 | 8 |
|---|---|---|---|---|---|---|---|---|
| **1. Gender** | — | | | | | | | |
| **2. Info. Preference** | 0.0554* | — | | | | | | |
| **3. Support** | 0.0546* | -0.1105** | — | | | | | |
| **4. SOM** | -0.0380* | -0.0834** | -0.1689** | — | | | | |
| **5. DEP** | -0.0016 | -0.0366* | -0.2535** | 0.7801** | — | | | |
| **6. ANX** | 0.0046 | -0.0731** | -0.1694** | 0.7962** | 0.8490** | — | | |
| **7. GSI** | 0.0032 | -0.0483* | -0.2181** | 0.8648** | 0.9543** | 0.9285** | — | |
| **8. BRIAN** | 0.0517* | 0.0515* | -0.2381** | 0.6212** | 0.6882** | 0.6491**** | 0.7017** | — |

Note

(1) **Support:** Social Support Total Scores; **SOM:** Somatization Scores; **DEP:** Depression Scores; **ANX:** Anxiety Scores; **GSI:** Global Severity Index; **BRIAN:** BRIAN Total Scores.

(2)

*: p < 0.0

**: p < 0.001.

**Table 3. Path analysis steps with fit indices.**

| Model | $X^2$ | p value* | RMSEA* | CFI* | TLI* | SRMR* |
|---|---|---|---|---|---|---|
| **1. Hypothesized Model** | 0.0000 | — | 0.0000 | 1.0000 | 1.0000 | 0.0000 |
| **2. Support→SOM*** | 0.5190 | 0.4710 | <0.001 | 1.0000 | 1.0020 | 0.0020 |
| **3. Info→Biorhythm*** | 0.0250 | 0.8740 | <0.001 | 1.0000 | 1.0040 | 0.0010 |
| **4. Gender→Biorhythm*** | 7.4650 | 0.0060 | 0.045 | 0.9970 | 0.9720 | 0.0150 |
| **5. Modified Model** | 7.9800 | 0.0463 | 0.0230 | 0.9980 | 0.9930 | 0.0150 |

Note:

**p value**: Chi $^2$ Test for model vs. saturated; **RMSEA**: Root mean squared error of approximation.

**CFI**: Comparative fit index; **TLI**: Tucker-Lewis index; **SRMR**: Standardized root mean squared residual.

**Support→SOM**: Removing pathway between social support and somatization scores, same as **Info→Biorhythm** and **Gender→Biorhythm.**

all study variables except BRIAN total scores. In addition, social support scores showed a negative correlation with both psychological distress scores and BRIAN total scores.

## Path analysis of the hypothesized model

According to correlations among study variables, we built a SEM to explore the overall relationship among the multiple study variables, and the fit indices were depicted (Table 3). The initial hypothesized path model fit the data poorly, and $X^2$ could not be calculated. Therefore, we had to sequentially remove three original pathways to generate the modified model (Table 3). Model 2 and Model 3 were invalid for Tucker-Lewis index more than 1. Moreover, although Model 4 fit all indices all in a reasonable range, p values were insignificant in two pathways after subsequent direct effects analysis (Table 4).

The modified model had good fit indices (Fig 1). Gender had a direct influence on social support, information preference, and somatization scores of psychological distress. Besides, somatization scores can be directly predicted by gender, information preference, and biological rhythm. In addition, it can be indirectly predicted by social support.

**Influence of gender on endogenous variables and outcome variable.** Gender difference had direct associations with social support, information preference, and somatization symptoms of psychological distress. Women were associated with a higher level of social support (standardized, β=0.0590, p=0.001) and lower somatization scores of psychological distress (standardized β=-0.0872, p<0.001). Men (standardized β=0.0617, p<0.001) were associated

**Table 4. Direct effects in Model 4.**

| Path | Coefficient* | SE | Z value | p value | 95% CI |
|---|---|---|---|---|---|
| **Gender→Support** | 0.0590 | 0.0175 | 3.37 | <0.001 | 0.0247~0.0933 |
| **Gender→Info** | 0.0617 | 0.0174 | 3.54 | <0.001 | 0.0276~0.0959 |
| **Support→Info** | -0.1075 | 0.0174 | -6.19 | <0.001 | -0.1416~-0.0735 |
| **Gender→BioRhythm** | | | No Path | | |
| **Info→BioRhythm** | 0.0002 | 0.0172 | 0.01 | 0.9920 | -0.0335~0.0338 |
| **Support→BioRhythm** | -0.2348 | 0.0167 | -14.06 | <0.001 | -0.2675~-0.2020 |
| **Gender→SOM** | -0.0865 | 0.0127 | -6.82 | <0.001 | -0.1114~-0.0616 |
| **BioRhythm→SOM** | 0.6685 | 0.0101 | 66.16 | <0.001 | 0.6487~0.6884 |
| **Info→SOM** | -0.1468 | 0.0128 | -11.47 | <0.001 | -0.1718~-0.1217 |
| **Support→SOM** | -0.0095 | 0.0134 | -0.71 | 0.4794 | -0.0358~0.0168 |

Note: * standardized coefficients.

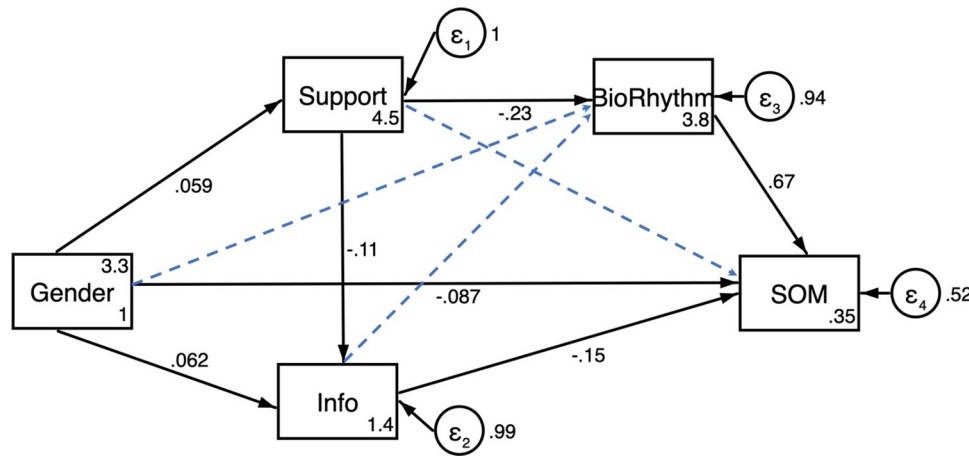

**Fig 1. Modified structural equation model.** Standardized beta coefficients are noted above each path. Solid lines indicate significant pathways, and perforated lines represent pathways removed from hypothesized model. Model fit indices: $X^2$ = 7.98 (p= 0.0463), Tucker-Lewis index = 0.9930, comparative fit index = 0.9980, root-mean-square error of approximation = 0.0230, and standardized root-mean-square residual = 0.0150, $R^2$=0.2142.

with more extreme media information. Neither gender nor information preference had any direct influence on biological rhythm.

**Influence of endogenous variables on outcome variable.** Social support was directly associated with biological rhythm (standardized β=-0.2348, p<0.001) and information preference (standardized β=-0.1075, p<0.001). The influence of social support on somatization symptoms was exerted through the process variable of biological rhythm. Lower biological rhythm total scores were associated with a lower level of somatization symptoms of psychological distress (standardized β=0.6707, p<0.001). For example, men might predict a lower level of social support, while weaker social support was associated with biological rhythm disorders, which further predicted higher scores of somatization symptoms. Besides, women might prefer to choose both negative and positive information, while men preferred more extreme information instead (Table 5).

## Discussion

In this cross-sectional study involving 3,237 participants, we found significant differences in social support total scores and global severity index between men and women, which goes in

**Table 5. Direct effects and indirect effects in modified model.**

| Path | Direct Effects | | | | | Indirect Effects | | | | |
|---|---|---|---|---|---|---|---|---|---|---|
| | Coefficient* | SE | Z value | p Value | 95% CI | Coefficient* | SE | Z value | p Value | 95% CI |
| **Gender→Support** | 0.0590 | 0.0175 | 3.36 | 0.001 | 0.2467~0.0933 | No Path | | | | |
| **Gender→Info** | 0.0617 | 0.0174 | 3.53 | <0.001 | 0.0276~0.0959 | -0.0063 | 0.0021 | -2.95 | 0.003 | -0.0105~-0.0021 |
| **Support→Info** | -0.1075 | 0.0174 | -6.15 | <0.001 | -0.1416~-0.0735 | No Path | | | | |
| **Gender→BioRhythm** | No Path | | | | | -0.0138 | 0.0042 | -3.26 | 0.001 | -0.0221~-0.0055 |
| **Support→BioRhythm** | -0.2348 | 0.0166 | -13.89 | <0.001 | -0.2673~-0.2022 | No Path | | | | |
| **Gender→SOM** | -0.0872 | 0.0127 | -6.85 | <0.001 | -0.1120~-0.0624 | -0.0174 | 0.0037 | -4.70 | <0.001 | -0.0247~-0.0101 |
| **BioRhythm→SOM** | 0.6707 | 0.0095 | 52.73 | <0.001 | 0.6521~0.6894 | No Path | | | | |
| **Info→SOM** | -0.1458 | 0.0127 | -11.45 | <0.001 | -0.1707~-0.1209 | No Path | | | | |
| **Support→SOM** | No Path | | | | | -0.1418 | 0.0122 | -11.64 | <0.001 | -0.1657~-0.1179 |

Note: * standardized coefficients.

line with the existing literature [35–37]. Nevertheless, most previous studies reported that women suffered from more somatization symptoms than men and were more vulnerable to psychological distress [36,38].

The most important objective of our study was to explain the possible associations of social support, information preference, and biological rhythm between gender and somatization symptoms. In our study, path analysis indicated that women had a higher level of social support, suggesting a smaller possibility of biological rhythm disorder and a lower level of somatization symptoms of psychological distress against the background of COVID-19 pandemic. The direct effect of gender difference on somatization symptoms was also statistically significant. This finding was partly in accordance with previous studies that confirmed the positive function of social support to relieve psychological distress, especially in chronic disease or traumatic natural disaster accidents [39–41]. In our model, we found woman was a protective factor, and this finding is inconsistent with other studies. Some extant studies showed that in women, the prevalence of psychological distress was higher and somatization symptoms were more obvious [10,38]. Nevertheless, a longitudinal study of the general population in China during COVID-19 pandemic suggested that men had a higher association with stress, anxiety, and depression than women [42]. Our findings may provide several explanations. First, women in general may access more easily sufficient social support from family members, collogues and / or friends [43,44]. This means women have more channels to obtain information about the COVID-19 pandemic. In addition, such communication can help them discern false, fake, or stigmatization epidemic information, which further relieves anxiety and somatization symptoms [45]. Second, for married men, home quarantine force them to stay with their family in a sense, which may very likely increase conflicts between the husband and wife due to limited recreational activities and personal space at home. In the context of the Chinese culture, most men are unwilling to tell their inner dissatisfaction to their intimate life partner, and it is difficult for them to express their inner anxious emotions, which might also increase their physical symptoms of psychological distress [46–48]. At the same time, Chinese fathers who have been largely absent in children's education [49] have to spend more time and energy in taking care of and educating their children during self-isolation due to COVID-19, which could also augment negative psychological feelings. Third, for unmarried or single men who live alone, self-isolation may be a big challenge because of insufficient social support and limited ways of expression. They may easily be confused by epidemic information and experience increased fear of COVID-19 pandemic. Therefore, enhancement of social support among men during COVID-19 home isolation is critical for alleviating their somatization symptoms of psychological distress [50].

In addition, our path analysis suggested that to alleviate somatization symptoms among men, we need to fortify their social support in addition to correcting their dysfunctioning biological rhythm such as insomnia, eating pattern disorder, or daily circadian social activities. Social support is not directly associated with somatization symptoms; and biological rhythm as a mediator plays an important role in relieving somatization symptoms of psychological distress. Our findings are in line with previous study results that psychological distress is associated with disruptions in sleep and circadian rhythm [51]. During home isolation people may stay up late unconsciously and have difficulty getting up the next morning, which could affect their daily eating pattern [52]. Furthermore, due to reduced outdoor physical activities, sleep problems and disturbance of the eating rhythm may also be aggravated. Biological rhythm disorder is a risk factor and needs to be intervened by professional psychologists or psychiatrists. Lockdown and home-quarantine restrict the access of psychological guidance. Hence, the internet cognitive behavioral therapy (i-CBT) as an effective measure can alleviate psychological distress and improve mental well-being, which is worthwhile to be implemented among those having insomnia or physical symptoms [53–55].

Finally, we found that preference of extreme information also increased somatization scores of psychological distress, and that men preferred to choose either negative or positive information. A chain mediation model study in Americans, Asians, and Europeans show that seeking for health information serves as a mediator between physical symptoms and the perceived influence of the COVID-19 pandemic. Overloading, conflicting, and ambiguous health information might increase burden of mental health [56]. Gender difference indeed affects public information preference and their extent of risk perception. Related health-seeking behavior could also be influenced by gender. Women are more capable of perceiving risks and thus will be more proactive in taking related health-seeking measures to weaken the negative impact of epidemic information [57,58]. Stronger social support in women can guarantee effective interpersonal information exchanges, which may contribute to confirming the reliability of epidemic information. Therefore, although gender difference has been observed in the access to epidemic information, the sufficient social support and varieties of communication channels that women could obtain partly compensate for the inadequacy in the access to media information. Therefore, women tend to be in a more neutral position when facing epidemic information of various kinds. These results have been partly confirmed by previous studies [16,18]. The preference for specific information may be key determinants of the individual's perception of risk regarding the COVID-19 pandemic. The "24-hour a day, 7 days a week" exposure to intensive and extensive media coverage of the COVID-19 pandemic amplifies risk perception and fear, making the general public anxious in the face of uncertainty. Such uncertainty greatly increases the individual's psychological burden [59]. Besides, the acceptance of COVID-19 vaccine was also affected by misleading, contradictory media information. To some extent, rumors and stigma for vaccine increase an individual's hesitancy for vaccination uptake. COVID-19 related somatic symptoms and other psychological distress are associated with higher willingness of vaccination uptake [60]. In particular, patients with mental illness have a higher vaccination acceptance [61,62]. In addition, individuals differ largely in risk perception by educational backgrounds and knowledge levels [63]. Therefore, as emphasized in previous studies, the dissemination of epidemic information should be dedicated to meeting the information needs of diverse sociodemographic and ethnic groups [18,64]. In addition, for different gender groups, the provision of epidemic information should also be tailored to individual needs.

## Limitations

The present study has three major limitations. First, because of the cross-sectional nature of the study, the relationships among study variables demonstrated in the structured model were based on strong theoretical rationales. Future research is needed to further employ longitudinal panel data to better understand causal inferences among gender, social support, information preference, biological rhythm, and psychological distress. Second, all scales used in the present study are self-rating questionnaires. Self-reported bias was thus inevitable due to personal attitudes. Finally, our sample may not be representative because most respondents did not come from high-risk areas like Hubei Province. The threat for COVID-19 and the corresponding psychological reaction may differ by risk areas.

## Conclusion

The study is the first step to uncover the direct and indirect effects of gender on somatization symptoms of psychological distress during the COVID-19 pandemic, while social support, biological rhythm, and information preference can be used as possible mediators. These findings highlight the gender difference in study variables during the COVID-19 pandemic and the

importance of social support in alleviating psychological distress and biological rhythm disorders. Moreover, the influence of information preference on somatization symptoms of psychological distress differs greatly by gender, and public health policy-makers and mass media need to provide better-targeted epidemic information to different individuals.

## Supporting information

**S1 Table. Effect size of social-demographic information, psychological distress, social support, biological rhythm, and media information preference.**
(DOCX)

**S1 Appendix. Chinese version informed consent.**
(PDF)

## Author Contributions

**Conceptualization:** Huanhuan Fan, Xiaohong Ma.

**Data curation:** Yikai Dou, Huanhuan Fan, Xiao Yang.

**Formal analysis:** Yikai Dou.

**Investigation:** Yikai Dou, Huanhuan Fan, Yue Du, Yu Wang, Min Wang, Zijian Zhang, Xiongwei Qi, Yuling Luo, Ruiqing Luo.

**Methodology:** Yikai Dou, Xiao Yang, Yu Wang, Yuling Luo.

**Project administration:** Xiao Yang.

**Resources:** Xiaohong Ma.

**Software:** Yue Du.

**Supervision:** Xiaohong Ma.

**Writing – original draft:** Yikai Dou.

**Writing – review & editing:** Xiaohong Ma.

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
