## [Decision Letter · Decision Letter 0]

7 Dec 2021

PONE-D-21-14840

Impact of the COVID-19 Pandemic on Psychological Distress and Biological Rhythm in China’s General Population: A Path Analysis Model

PLOS ONE

Dear Dr. Ma,

Thank you for submitting your manuscript to PLOS ONE. After careful consideration, we feel that it has merit but does not fully meet PLOS ONE’s publication criteria as it currently stands. Therefore, we invite you to submit a revised version of the manuscript that addresses the points raised during the review process.

Please address all reviewer comments. However, please note that you are not required to cite all references listed in the reviews. Please consider whether they are relevant to your manuscript; you may choose to cite them or not.

We look forward to receiving your revised manuscript.

Yours sincerely,

Yann Benetreau, PhD

Senior Editor

*PLOS ONE*

Journal Requirements:

3. Please complete all items on the Clinical Studies Checklist that are relevant for your submission, by following this link: http://journals.plos.org/plosone/s/file?id=dc11/PLOSOne_Clinical_Studies_Checklist.docx (Contact us at plosone@plos.org if you cannot access the document.) There may be overlap between the checklist items and other queries listed below; please address any duplicated queries both in your response email and on the checklist itself. Upload the completed Clinical Studies Checklist as file type “Other” when you re-submit your manuscript. This document is for internal journal use only and will not be published if your article is accepted. The requested information will help us to assess whether your submission complies with PLOS ONE’s policies and adheres to applicable reporting standards. Note that your manuscript may be rejected if you provide incomplete or inadequate responses to the checklist questions and that changing the ‘Section/Category’ of your article does not affect this requirement.

4. Please change "female” or "male" to "woman” or "man" as appropriate, when used as a noun (see for instance https://apastyle.apa.org/style-grammar-guidelines/bias-free-language/gender).

Reviewers' comments:

Reviewer's Responses to Questions

**Comments to the Author**

1. Is the manuscript technically sound, and do the data support the conclusions?

Reviewer #1: Yes

Reviewer #2: Yes

2. Has the statistical analysis been performed appropriately and rigorously? 

Reviewer #1: Yes

Reviewer #2: Yes

3. Have the authors made all data underlying the findings in their manuscript fully available?

Reviewer #1: Yes

Reviewer #2: No

4. Is the manuscript presented in an intelligible fashion and written in standard English?

Reviewer #1: Yes

Reviewer #2: Yes

5. Review Comments to the Author

Reviewer #1: I have the following comments for the authors to address. I am happy to review this paper again.

1) Under the Introduction, the authors stated "Social distancing, self-isolation and travel restrictions have led to a reduced workforce across all economic sectors and caused many jobs to be lost'". Please discuss the impact of social distancing, lockdown and facemask use on mental health:

Social distancing on mental health:

Impact of COVID-19 on Economic Well-Being and Quality of Life of the Vietnamese During the National Social Distancing. Front Psychol. 2020 Sep 11;11:565153. doi: 10.3389/fpsyg.2020.565153. PMID: 33041928; PMCID: PMC7518066.

Lockdown on mental health:

Anxiety and Depression Among People Under the Nationwide Partial Lockdown in Vietnam. Front Public Health. 2020;8:589359. Published 2020 Oct 29. doi:10.3389/fpubh.2020.589359

Facemask on mental health:

The Association Between Physical and Mental Health and Face Mask Use During the COVID-19 Pandemic: A Comparison of Two Countries With Different Views and Practices. Front Psychiatry. 2020;11:569981. Published 2020 Sep 9. doi:10.3389/fpsyt.2020.569981

2) Under the Introduction, the authors stated "Approximately 10%-30% general population were much concerned with being infected when an influenza outbreak occurred [10]." It is more appropriate to discuss the impact on general population during the pandemic instead of influenza. The following is a multinational study and its finding worth mentioning:

The impact of COVID-19 pandemic on physical and mental health of Asians: A study of seven middle-income countries in Asia. PLoS One. 2021 Feb 11;16(2):e0246824. doi: 10.1371/journal.pone.0246824. PMID: 33571297.

3) Under the discussion, the authors stated "Nevertheless, most previous studies reported that females suffer from more somatization symptoms than males and are more vulnerable to psychological distress [32, 34]." Reference 32 and 34 are not from China. Please check with studies from China whether they report similar or different finding:

A Longitudinal Study on the Mental Health of General Population during the COVID-19 Epidemic in China [published online ahead of print, 2020 Apr 13]. Brain Behav Immun. 2020; S0889-1591(20)30511-0. doi:10.1016/j.bbi.2020.04.028

4) Under the discussion, the authors should mention the relationship between physical or somatic symptoms and mental health based on the following study: "A chain mediation model on COVID-19 symptoms and mental health outcomes in Americans, Asians and Europeans. Sci Rep 11, 6481 (2021). " ext-link-type="uri" xlink:type="simple">https://doi.org/10.1038/s41598-021-85943-7"

5) Please discuss how to reduce somatic symptoms. I recommend Internet cognitive behavior therapy (iCBT) as it will reduce face to face contact and improve mental well being:

The most evidence-based treatment is cognitive behaviour therapy (CBT), especially Internet CBT that can prevent the spread of infection during the pandemic. Please refer to the following studies:

Use of Cognitive Behavior Therapy (CBT) to treat psychiatric symptoms during COVID-19:

Mental Health Strategies to Combat the Psychological Impact of COVID-19 Beyond Paranoia and Panic. Ann Acad Med Singapore. 2020;49(3):155‐160.

Cost-effectiveness of iCBT:

Moodle: The cost effective solution for internet cognitive behavioral therapy (I-CBT) interventions. Technol Health Care. 2017;25(1):163-165. doi: 10.3233/THC-161261. PMID: 27689560.

Internet CBT can treat psychiatric symptoms such as insomnia:

Efficacy of digital cognitive behavioural therapy for insomnia: a meta-analysis of randomised controlled trials. Sleep Med. 2020 Aug 26;75:315-325. doi: 10.1016/j.sleep.2020.08.020. Epub ahead of print. PMID: 32950013.

6) Please discuss how somatic symptoms may affect COVID-19 vaccine as people may think somatic symptoms as side effects of vaccines. Please comment on the following vaccine study and how somatic symptoms may affect attitude towards COVID-19 vaccine:

Attitudes toward COVID-19 vaccination and willingness to pay: Comparison of people with and without mental disorders in China. BJPsych Open, 7(5), E146. doi:10.1192/bjo.2021.979

Reviewer #2: The article is well-written and provides some interesting insight on the psychological impact of COVID-19, with a particular attention on gender differences.

The references are provided to a satisfying extent. Finally, the findings support the authors' conclusions and the sample size is totally reliable.

I just have a few concerns as regards the SEM/Path analysis. I wasn't able to grasp whether the analysis at hand is a proper SEM (therefore, with at least 1 latent variable) or a mere path analysis wherein all the variables are observed and we can assume no error. There seems to be some confusion about this aspect and I would definitely recommend further specifications on this issue.

In my opinion, this is the most critical point of the article; thus, it should be taken in serious consideration by the authors.

One last point: I am wondering whether the authors have considered to perform a multigroup SEM to see how the paths work for the two genders separately; i.e., whether some of the paths are significantly stronger for one of the genders.

I hereby attach a pdf file with 19 comments on several other points of the article.

6. PLOS authors have the option to publish the peer review history of their article (what does this mean?). If published, this will include your full peer review and any attached files.

Reviewer #1: No

Reviewer #2: No

---

## [Author Response · Author response to Decision Letter 0]

20 Jan 2022

Dear editor. Yann Benetreau,  

Re. Resubmission of PONE-D-21-14840, “Impact of the COVID-19 Pandemic on Psychological Distress and Biological Rhythm in China’s General Population: A Path Analysis Model”

Thank you for your action letter with the enclosed reviewers’ comments on our paper and your invitation to resubmit. We are grateful for the reviewers’ constructive comments. Our point-by-point response to the reviewers’ comments is immediately below this letter, and we have highlighted changes in red in the revised manuscript. We hope that by addressing the issues raised by the reviewers, the revised manuscript has been significantly improved and acceptable for publication in PLOS ONE. Should further revision be needed, please let us know. 

Thank you very much for your time and consideration. We are looking forward to hearing from you.

Sincerely,

Xiaohong Ma

---

## [Decision Letter · Decision Letter 1]

18 Apr 2022

PONE-D-21-14840R1Impact of the COVID-19 Pandemic on Psychological Distress and Biological Rhythm in China’s General Population: A Path Analysis ModelPLOS ONE

Dear Dr. Ma,

Thank you for submitting your manuscript to PLOS ONE. After careful consideration, we feel that it has merit but does not fully meet PLOS ONE’s publication criteria as it currently stands. Therefore, we invite you to submit a revised version of the manuscript that addresses the points raised during the review process. Academic Editor: Thank you for your revision. One of the reviewers found minor issues that need to be fixed before the paper is published. Please address them and resubmit. 

We look forward to receiving your revised manuscript.

Kind regards,

Abraham Salinas-Miranda, MD, PhD

Academic Editor

PLOS ONE

Journal Requirements:

Reviewers' comments:

Reviewer's Responses to Questions

**Comments to the Author**

1. If the authors have adequately addressed your comments raised in a previous round of review and you feel that this manuscript is now acceptable for publication, you may indicate that here to bypass the “Comments to the Author” section, enter your conflict of interest statement in the “Confidential to Editor” section, and submit your "Accept" recommendation.

Reviewer #1: All comments have been addressed

Reviewer #3: (No Response)

2. Is the manuscript technically sound, and do the data support the conclusions?

Reviewer #1: Yes

Reviewer #3: Yes

3. Has the statistical analysis been performed appropriately and rigorously? 

Reviewer #1: Yes

Reviewer #3: Yes

4. Have the authors made all data underlying the findings in their manuscript fully available?

Reviewer #1: Yes

Reviewer #3: (No Response)

5. Is the manuscript presented in an intelligible fashion and written in standard English?

Reviewer #1: Yes

Reviewer #3: (No Response)

6. Review Comments to the Author

Reviewer #1: I recommend publication and thanks for your amendments. This is a very good paper to be published and benefits the academia.

Reviewer #3: The authors sought to assess the influence of gender on social support, information preference, biological rhythm, and psychological distress during the COVID-19 pandemic. This addresses the gap of limited studies on the influence of gender on psychosomatic outcomes in the context of the COVID-19 pandemic. The authors used path analysis to assess the primary outcome of somatization. Overall, they found gender differences in factors predicting somatization and the important role of social support as a protective factor. Because of the paucity of literature on the influence of gender on somatization in the Chinese population in the context of COVID-19, this study helps to address this literature gap.

Overall

• This is a sound paper however there are numerous grammatical errors and areas where the authors sometimes switch between tenses (i.e. switch between present tense and past tense) that will need to be addressed.

• Standardized coefficients are not presented.

Introduction

1. Since you cited Chinese studies, the sentence on influenza is not needed: “Approximately 10%-30% general population were much concerned with being infected when there was an influenza outbreak[11]”

2. Typos in this sentence: “Moreover, a survey based on seven middle income countries in the general population showed high risk factors on mental health during the COVID-19 pandemic which related to single or separated status, high education level, and age 30 years [12].”

3. This sentence is confusing: “Furthermore, social media of unreliable sources usually provide much ambiguous epidemic information, and overloaded information may cause psychological distress in turn [16]. For example, while facemask wearing is a positive precaution, it still goes through social stigma, arousing mixed opinion and contradictory messaging from the media, which all lead to public fear and confusion [17].” Should it read like this instead? “Furthermore, social media using unreliable sources, usually provide ambiguous epidemic information, and information overload may cause psychological distress in turn [16]. For example, while facemask wearing is a positive precaution, its use brings about social stigma, arousing mixed opinion and contradictory messaging from the media, which all lead to public fear and confusion [17].” Please clarify.

4. Remove “etc” from the sentence: “Staying up late, getting up late, and lying in bed during non-sleeping …, social activities, etc. and aggravating people's physical and psychological distress in the meantime.”

Methods

1. Typos with APPS (make it lowercase) and ‘research’ is plural: “2) Both negative and positive media information. This question was designed based on some previous research which aimed to reflect the preference of different respondents for the magnanimity of media information on cellphone social apps or television[16, 20, 32, 33].”

2. The authors mention aiming to study possible causal relationships. However, causality cannot be assessed in a cross-sectional study. Please revise. Also correction with grammar: “ Finally, aiming to study the possible causal relationship…path analysis to test the relationship among interrelated study variables in a hypothesized model.”

Results

1. For Chi square the authors use both lower “x2” and upper case “X2”. Please be consistent and use upper case X2 throughout manuscript. Similar suggestion for “p values”. Use P or p to match how it is recommended by the journal. ‘p’ for values of p0.001.

2. Correction: (S1 Table shows the effect size of study variables…)

3. The authors go back and forth between using “men”, “women”, “males”, and “females”. Please be consistent and use terms appropriately.

4. It is suggested to remove the word “obviously” from the manuscript.

5. Again, the mention of exploring causality is mentioned and is not possible for a cross-sectional study.

6. Please report standardized coefficients and R2 values for the final model in table.

Discussion

1. Did the authors mean “In this cross-sectional study involving 3,237 participants, we found significant differences in social support total scores and…”?

2. Again, the authors switch between using “men”, “women”, “males”, and “females”. Please address this.

3. Correction: “Third,for unmarried or single men who lived alone, self-isolation may be a big challenge…be confused by epidemic information and experience increased fear of the COVID-19 pandemic.”

4. Correction: “Besides, the acceptance of the COVID-19 vaccine is also affected by…”

5. This sentence is confusing: “To some extent, hesitation in front of vaccine can be increased by rumors and stigmas.” Are the authors trying to say that rumors and stigma increase an individual’s hesitancy for vaccination uptake? Please reword for clarity.

6. Please use correct in-text citation based on journal style to cite: “Therefore, as Wong L P emphasized, the dissemination of epidemic information…”

7. PLOS authors have the option to publish the peer review history of their article (what does this mean?). If published, this will include your full peer review and any attached files.

Reviewer #1: No

Reviewer #3: No

---

## [Author Response · Author response to Decision Letter 1]

2 Jun 2022

Dear Reviewers,

We apologize for the grammatical errors that still persist. We have read the comments carefully and prepared one-by-one replies in Response to Reviewers. And we hope this version is better and has addressed your concern regarding language. All the changes were marked in red in Revised Manuscript with Track Changes. 

Thank you very much for your time and consideration. We are looking forward to hearing from you.

Sincerely,

Xiaohong Ma

---

## [Editor Report · Decision Letter 2]

28 Jun 2022

Impact of the COVID-19 Pandemic on Psychological Distress and Biological Rhythm in China’s General Population: A Path Analysis Model

PONE-D-21-14840R2

Dear Dr. Ma,

We’re pleased to inform you that your manuscript has been judged scientifically suitable for publication and will be formally accepted for publication once it meets all outstanding technical requirements.

Kind regards,

Abraham Salinas-Miranda, MD, PhD

Academic Editor

PLOS ONE

Additional Editor Comments (optional):

As Academic Editor, I have found the authors' responses adequate for the reviewers' comments. My recommendation is to accept for publication.
---

## [Editor Report · Acceptance letter]

1 Jul 2022

PONE-D-21-14840R2 

Impact of the COVID-19 pandemic on psychological distress and biological rhythm in China’s general population: A path analysis model 

Dear Dr. Ma:

I'm pleased to inform you that your manuscript has been deemed suitable for publication in PLOS ONE. Congratulations! Your manuscript is now with our production department. 

Kind regards, 

on behalf of

Dr. Abraham Salinas-Miranda 

Academic Editor

PLOS ONE